# Answering Science Exam Questions Using
# Query Reformulation with Background Knowledge

**Ryan Musa**[†]**, Xiaoyan Wang**[§]**, Achille Fokoue**[†]**, Nicholas Mattei**[*]**, Maria Chang**[†]**,
Pavan Kapanipathi**[†]**, Bassem Makni**[†]**, Kartik Talamadupula**[†]**, Michael Witbrock**[†]

[†]**IBM Research**
*IBM T.J. Watson Research Center*
*Yorktown Heights, NY 10598 USA*
{RAMUSA, ACHILLE, KAPANIPA, KRTALAMAD, WITBROCK}@US.IBM.COM
{MARIA.CHANG, BASSEM.MAKNI}@IBM.COM

[§] **University of Illinois at Urbana-Champaign**
*Department of Computer Science*
*Urbana, IL 61801 USA*
XIAOYAN5@ILLINOIS.EDU

[*] **Tulane University**
*Department of Computer Science*
*New Orleans, LA 70115 USA*
NSMATTEI@TULANE.EDU

## Abstract

Open-domain question answering (QA) is an important problem in AI and NLP that is emerging as a bellwether for progress on the generalizability of AI methods and techniques. Much of the progress in open-domain QA systems has been realized through advances in information retrieval (IR) methods and corpus construction. In this paper, we focus on the recently introduced ARC Challenge dataset, which contains 2,590 multiple choice questions authored for grade-school science exams. These questions are selected to be the most challenging for current QA systems, and current state of the art performance is only slightly better than random chance. We present a system that reformulates a given question into queries that are used to retrieve supporting text from a large corpus of science-related text. Our rewriter is able to incorporate background knowledge from ConceptNet. In tandem with a generic textual entailment system trained on SciTail that identifies support in the retrieved results, our system outperforms several strong baselines on the end-to-end QA task despite only being trained to identify essential terms in the original source question. We use a generalizable decision methodology – over the retrieved evidence and answer candidates – to select the best answer. By combining query reformulation, background knowledge, and textual entailment, our system is able to outperform several strong baselines on the ARC dataset.

## 1. Introduction

The recently released AI2 Reasoning Challenge (ARC) and accompanying ARC Corpus [Clark et al., 2018] is an ambitious test for AI systems that perform open-domain question answering (QA). This dataset consists of 2590 multiple choice questions authored for grade-school science exams; the questions are partitioned into an *Easy* set and a *Challenge* set. The Challenge set comprises questions that cannot be answered correctly by either a Pointwise Mutual Information (PMI-based)

solver, or by an Information Retrieval (IR-based) solver. Clark et al. [2018] also note that the simple information retrieval (IR) methodology (Elasticsearch) that they use is a key weakness of current systems, and conjecture that 95% of the questions can be answered using ARC corpus sentences.

ARC has proved to be a difficult dataset to perform well on, particularly its Challenge partition: existing systems like KG$^2$ [Zhang et al., 2018] achieve 31.70% accuracy on the test partition. Older models such as DecompAttn [Parikh et al., 2016] and BiDAF [Seo et al., 2017] that have shown good performance on other datasets – e.g. SQuAD [Rajpurkar et al., 2016] – perform only 1-2% above random chance.[1] The seeming intractability of the ARC Challenge dataset has only very recently shown signs of yielding, with the newest techniques attaining an accuracy of 42.32% on the Challenge set [Sun et al., 2018].[2]

An important avenue of attack on ARC was identified in Boratko et al. [2018a,b], which examined the knowledge and reasoning requirements for answering questions in the ARC dataset. The authors note that "*simple reformulations to the query can greatly increase the quality of the retrieved sentences*". They quantitatively measure the effectiveness of such an approach by demonstrating a 42% increase in score on ARC-Easy using a pre-trained version of the DrQA model [Chen et al., 2017]. Another recent tack that many top-performing systems for ARC have taken is the use of natural language inference (NLI) models to answer the questions [Zhang et al., 2018, Khot et al., 2018]. The NLI task – also sometimes known as recognizing textual entailment – is to determine whether a given natural language *hypothesis h* can be inferred from a natural language *premise p*. The NLI problem is often cast as a classification problem: given a hypothesis and premise, classify their relationship as either *entailment*, *contradiction*, or *neutral*. NLI models have improved state of the art performance on a number of important NLP tasks [Yin et al., 2018, Parikh et al., 2016, Chen et al., 2018] and have gained recent popularity due to the release of large datasets [Bowman et al., 2015, Khot et al., 2018, Williams et al., 2018, Wang et al., 2018b]. In addition to the NLI models, other techniques applied to ARC include using pre-trained graph embeddings to capture commonsense relations between concepts [Zhong et al., 2018]; as well as the current state-of-the-art approach that recasts multiple choice question answering as a reading comprehension problem that can also be used to fine-tune a pre-trained language model [Sun et al., 2018].

ARC Challenge represents a unique obstacle in the open domain QA world, as the questions are specifically selected to *not* be answerable by merely using basic techniques augmented with a high quality corpus. Our approach combines current best practices: it retrieves highly salient evidence, and then judges this evidence using a general NLI model. While other recent systems for ARC have taken a similar approach [Ni et al., 2018, Mihaylov et al., 2018], our extensive analysis of both the rewriter module as well as our decision rules sheds new light on this unique dataset.

In order to overcome some of the limitations of existing retrieval-based systems on ARC and other similar corpora, we present an approach that uses the original question to produce a set of reformulations. These reformulations are then used to retrieve additional supporting text which can then be used to arrive at the correct answer. We couple this with a textual entailment model and a robust decision rule to achieve good performance on the ARC dataset. We discuss important lessons learned in the construction of this system, and key issues that need to be addressed in order to move forward on the ARC dataset.

---

1. http://data.allenai.org/arc/
2. https://leaderboard.allenai.org/arc/submissions/public

## 2. Related Work

Teaching machines how to read, reason, and answer questions over natural language questions is a long-standing area of research; doing this well has been a very important mission of both the NLP and AI communities. The Watson project [Ferrucci et al., 2010] – also known as `DeepQA` – is perhaps the most famous example of a question answering system to date. That project involved largely factoid-based questions, and much of its success can be attributed to the quality of the corpus and the NLP tools employed for question understanding. In this section, we look at the most relevant prior work in improving open-domain question answering.

### 2.1 Datasets

A number of datasets have been proposed for reading comprehension and question answering. Hirschman et al. [1999] manually created a dataset of 3rd and 6th grade reading comprehension questions with short answers. The techniques that were explored for this dataset included pattern matching, rules, and logistic regression. MCTest [Richardson et al., 2013] is a crowdsourced dataset, and comprises of 660 elementary-level children's fictional stories, which are the source of questions and multiple choice answers. Questions and answers were constructed with a restricted vocabulary that a 7 year-old could understand. Half of the questions required the answer to be derived from two sentences, with the motivation being to encourage research in multi-hop (one-hop) reasoning. Recent techniques such as those presented by Wang et al. [2015] and Yin et al. [2016] have performed well on this dataset.

The original SQuAD dataset [Rajpurkar et al., 2016] quickly became one of the most popular datasets for reading comprehension: it uses Wikipedia passages as its source, and question-answer pairs are created using crowdsourcing. While it is stated that SQuAD requires logical reasoning, the complexity of reasoning required is far lesser than that required by the AI2 standardized tests dataset [Clark and Etzioni, 2016, Kembhavi et al., 2017]. Some approaches have already attained human-level performance on the first version of SQuAD. More recently, an extended version of SQuAD was released that includes over 50,000 additional questions where the answer cannot be found in source passages [Rajpurkar et al., 2018]. While unanswerable questions in SQuAD 2.0 add a significant challenge, the answerable questions are the same (and have the same reasoning complexity) as the questions in the first version of SQuAD. NewsQA [Trischler et al., 2016] is another dataset that was created using crowdsourcing; it utilizes passages from $10,000$ news articles to create questions.

Most of the datasets mentioned above are primarily closed world/domain: the answer exists in a given snippet of text that is provided to the system along with the question. On the other hand, in the open domain setting, the question-answer datasets are constructed to encompass the whole pipeline for question answering, starting with the retrieval of relevant documents. SearchQA [Dunn et al., 2017] is an effort to create such a dataset; it contains 140K question-answer (QA) pairs. While the motivation was to create an open domain dataset, SearchQA provides text that contains 'evidence' (a set of annotated search results) and hence falls short of being a complete open domain QA dataset. TriviaQA [Joshi et al., 2017] is another reading comprehension dataset that contains 650K QA pairs with evidence.

Datasets created from standardized science tests are particularly important because they include questions that require complex reasoning techniques to solve. A survey of the knowledge base requirements for answering questions from early science questions was performed by Clark et al.

[2013]. The authors concluded that advanced inference methods were necessary for many of the questions, as they could not be answered by simple fact based retrieval. Partially resulting from that analysis, a number of science-question focused datasets have been released over the past few years. The AI2 Science Questions dataset was introduced by Clark [2015] along with the Aristo Framework, which we build off of. This dataset contains over 1,000 multiple choice questions from state and federal science questions for elementary and middle school students.[3] The SciQ dataset [Welbl et al., 2017] contains 13,679 crowdsourced multiple choice science questions. To construct this dataset, workers were shown a passage and asked to construct a question along with correct and incorrect answer options. The dataset contained both the source passage as well as the question and answer options.

## 2.2 Query Expansion & Reformulation

Query expansion and reformulation – particularly in the area of information retrieval (IR) – is well studied [Azad and Deepak, 2017]. The primary motivation for query expansion and reformulation in IR is that a query may be too short, ambiguous, or ill-formed to retrieve results that are relevant enough to satisfy the information needs of users. In such scenarios, query expansion and reformulation have played a crucial role by generating queries with (possibly) new terms and weights to retrieve relevant results from the IR engine. While there is a long history of research on query expansion [Maron and Kuhns, 1960], Rocchio's relevance feedback gave it a new beginning [Rocchio, 1971]. Query expansion has since been applied to many applications, such as Web Personalization, Information Filtering, and Multimedia IR. In this work, we focus on query expansion as applied to question answering systems, where paraphrase-based approaches using an induced semantic lexicon [Fader et al., 2013] and machine translation techniques [Dong et al., 2017] have performed well for both structured query generation and answer selection. Open-vocabulary reformulation using reinforcement learning has also been demonstrated to improve performance on factoid-based datasets like SearchQA, though increasing the fluency and variety of reformulated queries remains an ongoing effort [Buck et al., 2017].

## 2.3 Retrieval

Retrieving relevant documents/passages is one of the primary components of open domain question answering systems [Wang et al., 2018a]. Errors in this initial module are propagated down the line, and have a significant impact on the ultimate accuracy of QA systems. For example, the latest sentence corpus released by AI2 (i.e. the ARC corpus) is estimated by Clark et al. [2018] to contain the answers to 95% of the questions in the ARC dataset. However, even state of the art systems that are not completely IR-based (but use neural or structured representations) perform only slightly above chance on the Challenge set. This is at least partially due to early errors in passage retrieval. Recent work by Buck et al. [2018] and Wang et al. [2018a] have identified improving the retrieval modules as the key component in improving state of the art QA systems.

## 3. System Overview

Our overall pipeline is illustrated in Figure 1 and comprises three modules: the *Rewriter* reformulates a question into a set of queries; the *Retriever* uses those queries to obtain relevant passages

---

3. http://data.allenai.org/ai2-science-questions

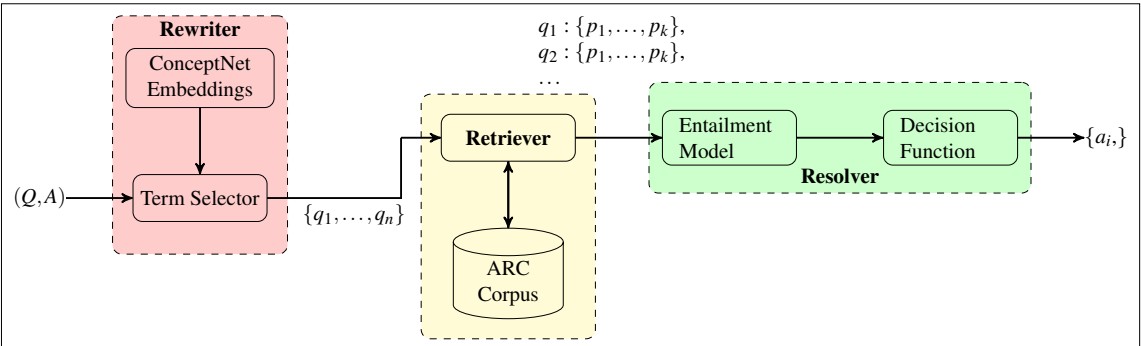

Figure 1: Our overall system architecture. The Rewriter module reformulates a natural-language question into queries by selecting salient terms. The Retriever module executes these queries to obtain a set of relevant passages. Using the passages as evidence, the Resolver module computes entailment probabilities for each answer and applies a decision function to determine the final answer set.

from a text corpus, and the *Resolver* uses the question and the retrieved passages to select the final answer(s).

More formally, a pair $(Q,A)$ composed of a question $Q$ with a set of answers $a_i \in A$ is passed into the Rewriter module. This module uses a term selector which (optionally) incorporates background knowledge in the form of embeddings trained using Knowledge Graphs such as ConceptNet to generate a set of reformulated queries $\mathcal{Q} = \{q_1, \ldots, q_n\}$. In our system, as with most other systems for ARC Challenge [Clark et al., 2018], for each question $Q$, we generate a set of queries where each query uses the same set of terms with one of the answers $a_i \in A$ appended to the end. This set of queries is then passed to a Retriever which issues the search over a corpus to retrieve a set of $k$ relevant passages *per query* to create a set of passages $P = \{q_1 p_1, \ldots, q_1 p_k, q_2 p_1, \ldots, q_n p_k\}$ that are passed to the Resolver. The Resolver contains two components: (1) the entailment model and (2) the decision function. We use match-LSTM [Wang and Jiang, 2016a] trained on SciTail [Khot et al., 2018] for our entailment model and for each passage passed in we compute the probability that each answer is entailed from the given passage and question. This information is passed to the decision function which selects a non-empty set of answers to return.

## 3.1 Rewriter Module

For the *Rewriter* module, we investigate and evaluate two different approaches to reformulate queries by retaining only their most salient terms: a sequence to sequence model similar to Sutskever et al. [2014] and models based on the recent work by Yang and Zhang [2018] on Neural Sequence Labeling. Figure 3 provides examples of the queries that are obtained by selecting terms from the original question using each of the models described in this section.

### 3.1.1 SEQ2SEQ MODEL FOR SELECTING RELEVANT TERMS

We first consider a simple sequence-to-sequence model shown in Figure 2 that translates a sequence of terms in an input query into a sequence of 0s and 1s of the same length. The input terms are passed to an encoder layer through an embedding layer initialized with pre-trained embeddings (e.g., GloVe [Pennington et al., 2014]). The outputs of the encoder layer are decoded, using an attention mechanism [Bahdanau et al., 2014], into the resulting sequence of 0s and 1s that is used

as a mask to select the most salient terms in the input query. Both the encoder and decoder layers are implemented with a single hidden bidirectional GRU layer ($h = 128$).

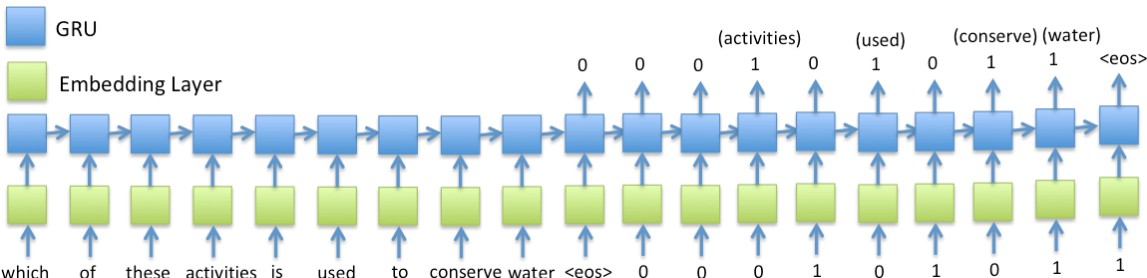

Figure 2: Seq2Seq query reformulation model. A sequence of terms from the original query is translated into a sequence of 0s and 1s which serves as a mask used to select the most salient terms.

### 3.1.2 NCRF++ MODELS FOR SELECTING RELEVANT TERMS

Our second approach to identifying salient terms comprises four models implemented with the NCRF++ sequence-labeling toolkit[4] of Yang and Zhang [2018]. Our basic `NCRF++` model uses a bi-directional LSTM with a single hidden layer ($h = 200$) where the input at each token is its 300-dimensional pre-trained GloVe embedding [Pennington et al., 2014]. Additional models incorporate background knowledge in the form of graph embeddings derived using the ConceptNet knowledge base [Speer et al., 2017] using three knowledge graph embedding approaches: `TransH` [Wang et al., 2014], `CompleX` [Trouillon et al., 2016], and the `PPMI` embeddings released with ConceptNet [Speer et al., 2017]. Entities are linked with the text by matching their surface form with phrases of up to three words. For each token in the question, we concatenate its word embedding with a 10-dimensional vector indicating whether the token is part of the surface form of a ConceptNet entity. We then append either the 300-dimensional vector corresponding to the embedding of that entity in ConceptNet, or a single randomly initialized `UNK` vector when a token is not linked to an entity. The final prediction is performed left-to-right using a CRF layer that takes into account the preceding label. We train the models for 50 iterations using SGD with a learning rate of 0.015 and learning rate decay of 0.05.

### 3.1.3 TRAINING AND EVALUATION OF REWRITER MODELS

Before integrating the rewriter module into our overall system (Figure 1), the two rewriter models (seq2seq and NCRF++) are first trained and tested on the Essential Terms dataset introduced by Khashabi et al. [2017].[5] This dataset consists of 2,223 crowd-sourced questions. Each word in a question is annotated with a numerical rating on the scale 1–5 that indicates the importance of the word.

Table 1 presents the results of our models evaluated on Essential Terms dataset along with those of two state-of-the-art systems: ET Classifier Khashabi et al. [2017] and ET Net [Ni et al., 2018]. ET Classifier trains an SVM using over 120 features based on the dependency parse, semantic features of the sentences, cluster representations of the words, and properties of question words. While the

---

4. https://github.com/jiesutd/NCRFpp
5. https://github.com/allenai/essential-terms

Emily made a cup of tea and stirred it with a spoon. The spoon became warm. How was the heat from the tea transferred to the spoon? (Correct answer: ***conduction***)

**seq2seq**:    tea stirred it spoon

**NCRF++**:    tea heat tea transferred

**TransH**:    tea stirred spoon became warm heat tea transferred spoon

**PPMI**:    heat tea transferred spoon

**CompleX**:    cup tea spoon heat transferred spoon

Figure 3: Example question and selected terms for each of our rewriter models.

| Method | Acc | Pr | Re | F1 |
|---|---|---|---|---|
| ET Classifier [Khashabi et al., 2017] | 0.75 | 0.91 | 0.71 | 0.80 |
| ET Net [Ni et al., 2018] | – | 0.74 | 0.90 | 0.81 |
| Seq2Seq - 6B.50d | 0.75 | 0.52 | 0.23 | 0.32 |
| Seq2Seq - 6B.100d | 0.76 | 0.54 | 0.46 | 0.50 |
| Seq2Seq - 840B.300d | 0.77 | 0.57 | 0.42 | 0.49 |
| NCRF++ | 0.88 | 0.73 | 0.80 | 0.77 |
| CompleX | 0.88 | 0.74 | 0.80 | 0.77 |
| TransH | 0.88 | 0.75 | 0.77 | 0.76 |
| PPMI | 0.87 | 0.77 | 0.72 | 0.75 |

Table 1: Essential terms classification performance as measured by token-level (Acc)uracy, (Pr)ecision, (Re)call, and F1 score. The results for ET Classifier reflect the 70/9/21 train/dev/test split reported in Khashabi et al. [2017]. We follow ET Net in using a random 80/10/10 train/dev/test split performed after filtering out questions that appear in the ARC dev/test sets.

ET Classifier was evaluated using a 79/9/21 train/dev/test split, we follow Ni et al. [2018] in using an 80/10/10 split and remove questions from the Essential Terms dataset that appear in the ARC dev/test partitions.

The key insights from this experimental evaluation are as follows:

- NCRF++ significantly outperforms the seq2seq model with respect to all evaluation metrics (see results with GloVe 840B.300d).

- NCRF++ is competitive with respect to ET Net and ET Classifier (without the heavy feature engineering of the latter system). It has significantly better accuracy and recall than ET Classifier although its F1-score is 3% inferior. When used with CompleX graph embeddings [Trouillon et al., 2016], it has the same precision as ET Net, but its F1-score is 4% less.

- Finally, while the results in Table 1 do not seem to support the need for using ConceptNet embeddings, we will see in the next section that, on ARC Challenge dev set, incorporating

outside knowledge significantly increases the quality of passages that are available for downstream reasoning.

## 4. Retriever Module

Retrieving and providing high quality passages to the Resolver module is an important step in ensuring the accuracy of the system. In our system, a set of queries $\mathcal{Q} = \{q_1, \ldots, q_n\}$ are sent to the Retriever, which then passes these queries along with a number of passages to the Resolver module. We use Elasticsearch [Gormley and Tong, 2015], a state-of-the-art text indexing system. We index on the ARC Corpus that is provided as part of the ARC Dataset. Clark et al. [2018] claim that this 14M-sentence corpus covers 95% of the questions in the ARC Challenge, while Boratko et al. [2018a,b] observe that the ARC corpus contains many relevant facts that are useful to solve the annotated questions from the ARC training set. An important direction for future work is augmenting the corpus with other search indices and sources of knowledge from which passages can be retrieved.

## 5. Resolver Module

Given the retrieved passages, the system still needs to select a particular answer out of the answer set $A$. In our system we divide this process into two components: the entailment module and the decision rule. In previous systems both of these components have been wrapped into one. Separating them allows us to study each of them individually, and make more informed design choices.

### 5.1 Entailment Modules

While reading comprehension models like BiDAF [Seo et al., 2017] have been adapted to the multiple-choice QA task by selecting a span in the passage obtained by concatenating several IR results into a larger passage, recent high-scoring systems on the ARC Leaderboard have relied on textual entailment models. In the approach pioneered by Khot et al. [2018], a multiple choice question is converted into an entailment problem wherein each IR result is a *premise*. The question is turned into a fill-in-the-blank statement using a set of handcrafted heuristics (e.g. replacing wh-words). For each candidate answer, a *hypothesis* is generated by inserting the answer into the blank and the model's probability that the *premise* entails this *hypothesis* becomes the answer's score.

We use match-LSTM [Wang and Jiang, 2016a,b] trained on SciTail [Khot et al., 2018] as our textual entailment model. We chose match-LSTM because: (1) multiple reading comprehension techniques have used match-LSTM as a important module in their overall architecture [Wang and Jiang, 2016a,b]; and (2) match-LSTM models trained on SciTail achieve an accuracy of 84% on test (88% on dev), outperforming other recent entailment models such as DeIsTe [Yin et al., 2018] and DGEM [Khot et al., 2018].

Match-LSTM consists of an attention layer and a matching layer. Given premise $P = (t_1^p, t_2^p, ..., t_K^p)$ and hypothesis $H = (t_1^h, t_2^h, ..., t_N^h)$ where $t_i^p$ and $t_j^h$ are embedding vectors of corresponding words in premise and hypothesis. A contextual representation of premise and hypothesis is generated by encoding their embedding vectors using bi-directional LSTMs. Let $\mathbf{p}_i$ and $\mathbf{h}_j$ be the contextual representation of the $i$-th word in the premise and the $j$-th word in the hypothesis computed using BiLSTMs over its embedding vectors. Then, an attention mechanism is used to determine the attention weighted representation of the $j$ word in the hypothesis as follows: $\mathbf{a_j} = \sum_{i=1}^{K} \alpha_{ij} \mathbf{p_i}$

where $\alpha_{ij} = \dfrac{\exp(e_{ij})}{\sum_{r=1}^{K} \exp(e_{rj})}$ and where $e_{ij} = \mathbf{p}_i \cdot \mathbf{h}_j$. The matcher layer is an $LSTM(m)$ where the input $m_j = [\mathbf{a_j}; \mathbf{h_j}]$ ($[;]$ is the concatenation operator). Finally, the max-pooling result over the hidden states $\{\mathbf{h_j^m}\}_{j=1:N}$ of the matcher is used for softmax classification.

## 5.2 Decision Rules

In the initial study of the ARC Dataset, Clark et al. [2018] convert many existing question answering and entailment systems to work with the particular format of the ARC dataset. One of the choices that they made during this conversion was to decide how the output of the entailment systems, which consist of a probability that a given hypothesis is entailed from a premise, are aggregated to arrive at a final answer selection. The rule used, which we call the AI2 Rule for comparison, is to take the top-8 passage by Elasticsearch score after pooling all queries for a given question. Each one of these queries has a specific $a_i$ that was associated with it due to the fact that all queries are of the format $Q + a_i$. For each of these top-8 passages, the entailment score of $a_i$ is recorded and the top entailment score is used to select an answer.

In our system we decided to make this decision rule not part of the particular entailment system but rather a completely separate module. The entailment system is responsible for measuring the entailment for each answer option for each of the retrieved passages and passing this information to the decision rule. One can compute a number of statistics and filters over this information and then arrive at one or more answer selections for the overall system.

In addition to the AI2 Rule described above, we also experiment filtering by Elasticsearch score per individual $Q + a_i$ query (rather than pooling scores across queries).[6] Referred to in the next section as the MaxEntail Top-$k$ rule, this decision function select the answer(s) that have the greatest entailment probability when considering the top-$k$ passages retained per query.

## 6. Empirical Evaluation

Our goal in this section is to evaluate the effect that our query reformulation and result filtering techniques have on overcoming the IR bottleneck in the open-domain question answering pipeline. In order to isolate those effects on the Retriever module, it is imperative to avoid overfitting to the ARC Challenge Training set. Thus, for all experiments the Resolver module uses the same match-LSTM model trained on SciTail as described in Section 5.1. In the Rewriter module, all query reformulation models are trained on the same Essential Terms data described in Section 3.1 and differ only in architecture (seq2seq vs. NCRF) and in the embedding technique used to encode background knowledge. Finally, we tune the hyperparameters of our decision rules (i.e. the number of Elasticsearch results passed to considered by the Resolver module) on the ARC Challenge dev set. Our results on the dev set for 12 of our models and two different decision rules are summarized in Figure 4 and Figure 5. The final results for test set are provided in Table 2.

---

6. Note that combining Elasticsearch scores *across* passages is not typically considered a good idea; according to the Elasticsearch Best Practices FAQ "... the only purpose of the relevance score is to sort the results of the current query in the correct order. You should not try to compare the relevance scores from different queries."

## 6.1 Dev Set

We first consider the important question of *how many passages to investigate per query*: we can compare and contrast Figure 4 (AI2 Rule) and Figure 5 (Max Entailment of top-K per answer) which varies the number of passages *k* that are considered. The most obvious difference is that the results show that max entailment of top-*k* is strictly a better rule overall for both the original and split hypothesis. In addition to the overall score, keeping the top-*k* results per answer results in a smoother curve that is more amenable to calibration on the dev partition.

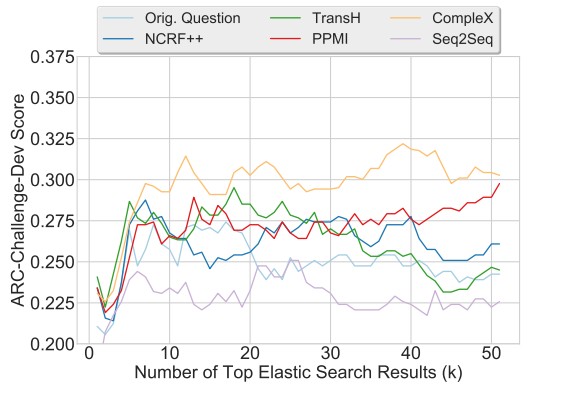

(a) AI2 Rule w/ original hypothesis

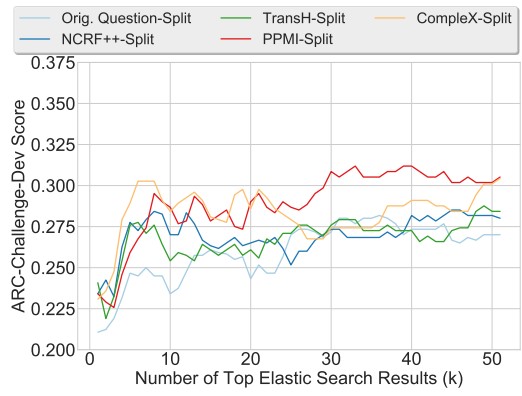

(b) AI2 Rule w/ split hypothesis

Figure 4: Performance of our models on the dev partition of the Challenge set using (a) the original hypothesis and (b) the split hypothesis as we vary the number of results *k* retained by AI2 Rule, i.e. overall Elasticsearch score.

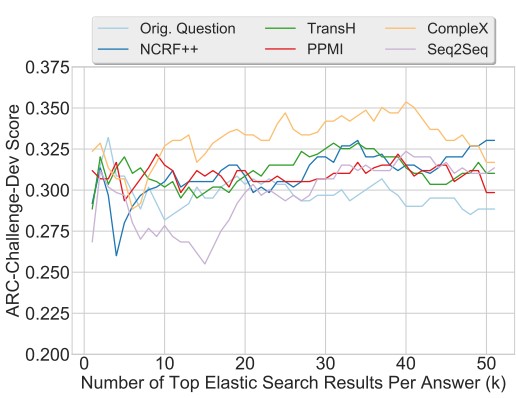

(a) MaxEntail Rule w/ original hypothesis.

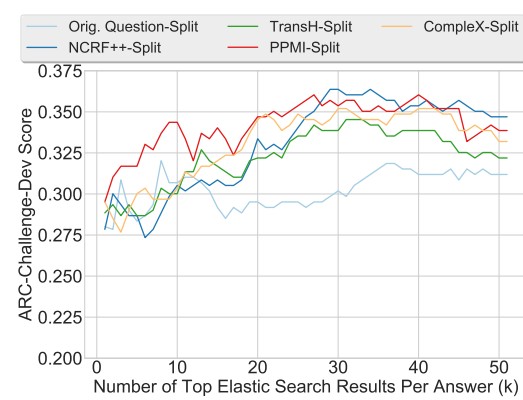

(b) MaxEntail Rule w/ split hypothesis.

Figure 5: Performance of our models on the dev partition of the Challenge set using (a) the original hypothesis and (b) the split hypothesis constructed by splitting a multi-sentence question as we vary the number of results *k* retained by the MaxEntail rule, i.e. Elasticsearch score per candidate answer.

Comparing sub-figures (a) and (b) in Figures 4 and 5, we find more empirical support for our decision to investigate splitting the hypothesis. The length of the questions in the Challenge versus Easy set average 21.8 v. 19.1 words, respectively; for the answers, the length is 4.9 words versus 3.8 respectively. One possible cause for poor performance on ARC Challenge is that entailment models are easily confused by very long, story based questions. Working off the annotations of Boratko

et al. [2018a,b], many of the questions of type "Question Logic" are of this nature. To address this, we "split" multi-sentence questions by (a) forming the hypothesis from only final sentence and (b) pre-pending the earlier sentences to each premise. Comparing across the figures we see that, in general, the modified hypothesis splitting leads to a small improvement in scores.

We also see the effect of including background knowledge via ConceptNet embeddings on the performance of the downstream reasoning task; this is particularly evident in Figure 5. All of the rewritten queries are superior to using the original question. Additionally, in both Figure 5 (a) and (b), the CompleX and PPMI embeddings are performing better than the base rewriter. This is a strong indication that using the background knowledge in specific ways can aid downstream reasoning tasks; this is contrary to the results of Mihaylov et al. [2018].

| Model | ARC-Challenge-Dev | | | ARC-Challenge-Test | | |
|---|---|---|---|---|---|---|
| | AI2 | Top-2 | Top-30 | AI2 | Top-2 | Top-30 |
| Orig. Question | 27.25 | 31.52 | 29.68 | 25.12 | 31.61 | 31.43 |
| Orig. Question-Split | 24.49 | 27.84 | 30.18 | 25.95 | 29.80 | 30.13 |
| Seq2Seq | 23.18 | 31.12 | 30.68 | 26.93 | 29.98 | 30.13 |
| NCRF++ | 27.59 | 31.35 | 32.02 | 26.26 | **33.18** | **33.20** |
| NCRF++-Split | 28.42 | 30.00 | **36.37** | 26.94 | 31.58 | 30.56 |
| TransH | 28.01 | 32.02 | 32.53 | 25.86 | 31.57 | 32.68 |
| TransH-Split | 27.59 | 29.34 | 33.86 | 25.77 | 30.06 | 31.58 |
| PPMI | 27.42 | 30.69 | 30.68 | **27.40** | 31.88 | 32.92 |
| PPMI-Split | 29.51 | 31.02 | 35.37 | 26.27 | 29.76 | 31.28 |
| CompleX | 29.59 | **32.86** | 34.20 | 26.44 | 30.20 | 31.66 |
| CompleX-Split | **30.26** | 28.51 | 35.20 | 26.74 | 29.93 | 31.54 |

Table 2: Results of our models on the dev/test partitions of the Challenge set when responding with the maximally entailed answer(s) based on a set of filtered Elasticsearch results. Answers are selected based on: the *AI2* rule over the top $k = 8$ individual results based on overall Elasticsearch score; (b) the MaxEntail rule over the *Top-2* results by Elasticsearch score per candidate answer (typically 8 results total); or (c) the MaxEntail rule retaining the *Top-30* results per answer (per Figure 5b).

## 6.2 Test Set

Considering the results on the dev set, we use the test set to evaluate the following decision rules for all of our system: the AI2 Rule, Top-2 per query, and Top-30 per query. We selected Top-2 as it is the closest analog to the AI2 Rule, and Top-30 because there is a clear and long peak from our initial testing on the dev set (per Figure 5b). The results of our run on test set can be found in Table 2. For the most direct comparison between the two methods (i.e., without splitting), all models using the Top-2 rule outperform the AI2 rule at at least 99.9% confidence using a paired t-test. We note that for the dev set, the split treatments nearly uniformly dominate the non-split treatments; while for the test set this is almost completely reversed (and for Original Question and PPMI, for which splitting outperforms non-splitting at 95% confidence). Perhaps more surprisingly, the more sophisticated ConceptNet embeddings are almost uniformly better on the dev set; while on the test set they are nearly uniformly worse. For context, we also provide the state of the ARC leaderboard at the time of submission with the addition to our top-performing system in Table 3.

| Model | ARC-Challenge Test | ARC-Easy Test |
|---|---|---|
| Reading Strategies [Sun et al., 2018] | 42.32 | 68.9 |
| ET-RR [Ni et al., 2018] | 36.36 | – |
| BiLSTM Max-Out [Mihaylov et al., 2018] | 33.87 | – |
| TriAN + f(dir)(cs) + f(ind)(cs) [Zhong et al., 2018] | 33.39 | – |
| **NCRF++/match-LSTM** | **33.20** | **52.22** |
| KG$^2$ [Zhang et al., 2018] | 31.70 | – |
| DGEM [Khot et al., 2018] | 27.11 | 58.97 |
| TableILP [Khashabi et al., 2016] | 26.97 | 36.15 |
| BiDAF [Seo et al., 2017] | 26.54 | 50.11 |
| DecompAtt [Parikh et al., 2016] | 24.34 | 58.27 |

Table 3: Comparison of our system with state-of-the-art systems for the ARC dataset. Numbers taken from ARC Leaderboard as of Nov. 18, 2018 Clark et al. [2018].

## 7. Discussion

Of the systems above ours on the leaderboard, only Ni et al. [2018] report their accuracy on both the dev set (43.29%) and the test set (36.36%). We suffer a similar loss in performance from 36.37% to 33.20%, demonstrating the risk of overfitting to a (relatively small) development set in the multiple-choice setting even when a model has few learnable parameters. As in this paper, Ni et al. [2018] pursue the approach suggested by Boratko et al. [2018a,b] in learning how to transform a natural-language question into a query for which an IR system can return a higher-quality selection of results. Both of these systems use entailment models similar to our match-LSTM [Wang and Jiang, 2016a] model, but also incorporate additional co-attention between questions, candidate answers, and the retrieved evidence.

Sun et al. [2018] present an an encouraging result for combating the IR bottleneck in open-domain QA. By concatenating the top-50 results of a single (joint) query and feeding the result into a neural reader optimized by several lightly-supervised 'reading strategies', they achieve an accuracy of 37.4% on the test set even without optimizing for single-answer selection. Integrating this approach with our query rewriting module is left for future work.

## 8. Conclusions and Future Work

In this paper, we present a system that answers science exam questions by retrieving supporting evidence from a large, noisy corpus on the basis of keywords extracted from the original query. By combining query rewriting, background knowledge, and textual entailment, our system is able to outperform several strong baselines on the ARC dataset. Our rewriter is able to incorporate background knowledge from ConceptNet and – in tandem with a generic entailment model trained on SciTail – achieves near state of the art performance on the end-to-end QA task despite only being trained to identify essential terms in the original source question.

There are a number of key takeaways from our work: first, researchers should be aware of the impact that Elasticsearch (or a similar tool) can have on the performance of their models. Answer candidates should not be discarded based on the relevance score of their top result; while (correct) answers are likely critical to retrieving relevant results, the original AI2 Rule is too aggressive in pruning candidates. Using an entailment model that is capable of leveraging background knowledge in a more principled way would likely help in filtering unproductive search results. Second, our

results corroborate those of Ni et al. [2018] and show that tuning to the dev partition of the Challenge set (299 questions) is extremely sensitive. Though we are unable to speculate on whether this is an artifact of the dataset or a more fundamental concern in multiple-choice QA, it is an important consideration for generating significant and reproducible improvements on the ARC dataset.

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
