# OpenReview forum: "Answering Science Exam Questions Using Query Reformulation with Background Knowledge"
_AKBC.ws/2019/Conference — AKBC 2019_

### Official Review · AnonReviewer1 · 2018-12-13
**The system is a combination of many existing techniques, and is outperformed by several works.**

**Rating:** 6
**Confidence:** 3

**Review:**

This paper introduces an end-to-end system to answer science exam questions for the ARC challenge. The system is a combination of several existing techniques, including (i) query rewriting based on seq2seq or NCRF++, (ii) answer retriever, (iii) entailment model based on match-LSTM, and (iv) knowledge graph embeddings. The description of the system is clear, and there is abundant ablation study. However, I have following concerns about this paper:

1. There seems to be no new techniques proposed in the system. Hence, the novelty of this work is questioned.
2. I do not understand why the authors use TransH, which is a KG embedding model that differentiates one entity into different relation-specific representations.
3. The system is significantly outperformed by Sun et al. 2018 and Ni et al. 2018.

---

> ### Author Response · Authors · 2019-02-01
> **Very incisive review**
>
> Thank you for the careful review of our paper.
>
> For (1) we must somewhat disagree -- since submission indeed there have been several new papers posted to arXiv which include rewriter modules for the ARC dataset inspired by the work of Boratko et al. (https://arxiv.org/pdf/1806.00358.pdf) but as of submission time the notion of combining a rewriter with the entailment module for ARC is novel.  We also point to Reviewer2’s comment, “Query reformulation methods have been used on several QA tasks (like Buck et al 2018 above), and incorporating background knowledge has been used before too (as described in the paper), but I think it’s fairly original to do both in the same time.”
>
> With respect to (2) -- we used TransH to have a common and easily trainable baseline for comparison. We do not have a priori beliefs about which aspects of an entity’s relations are useful for answering a particular science question, but given that many science questions pertain to different functional properties of matter it does not seem unreasonable to use the vector embedding provided by TransH to do so.
>
> With respect to (3) -- we have not yet incorporated transformer networks/BERT as done by Sun 2018 and Ni 2018 and some other groups have done (https://leaderboard.allenai.org/arc/submissions/public).  However, these papers have been posted between the time of submission to AKBC and the response period and we are working to integrate these aspects into future work. There's no question that massively pre-trained models like BERT represent a substantial shift in the state-of-the-art of NLP, and we expect that all future work in the near future should use them to represent text. However, we feel that the contributions of our paper are mainly focused around the end-to-end pipeline and providing techniques for leveraging a corpus of background knowledge that improve on the assumptions of prior work from e.g. AI2.

---

### Official Review · AnonReviewer3 · 2019-01-06
**good qa paper**

**Rating:** 6
**Confidence:** 4

**Review:**

This paper focuses on the recently introduced ARC Challenge dataset, which contains 2,590 multiple choice questions authored for grade-school science exams. The paper presents a system that reformulates a given question into queries that are used to retrieve supporting text from a large corpus of science-related text. The rewriter is able to incorporate background knowledge from ConceptNet. A textual entailment system trained on SciTail that identifies support in the retrieved results.  Experiments show that the proposed system is able to outperform several baselines on ARC.

* (Sec 2.2)  "[acl-2013] Paraphrase-driven learning for open question answering" and "[emnlp-2017] Learning to Paraphrase for Question Answering" can be added in the related work section.
* (Sec 3.1) Seq2seq predicts 0 and 1 to indicate whether the word is salient. A more straightforward method is using a pointer network for the decoder, which directly selects words from the input. This method should be more effective than seq2seq used in Sec 3.1.1.
* (Sec 3.1) How about the performance of removing the top crf layer? The LSTM layer and the classifier should play the most important role.
* How to better utilize external resources is an interesting topic and is potentially helpful to improve the results of answering science exam questions. For example, the entailment module described in Sec 5.1 can be trained on other larger data, which in turn helps the problem with smaller data. I would like to see more details about this.
* Are the improvements significant compared to the baseline methods? Significance test is necessary because the dataset is quite small.
* Experiments on large-scale datasets are encouraged.

---

> ### Author Response · Authors · 2019-02-01
> **Added example, significance tests, additional references & discussion**
>
> We very much appreciate the thoughtful suggestions of your review, and have added the citations for the paraphrase-based approaches to query reformulation in open QA to the related work section in our revision.
>
> The comments about pointer networks vs. our seq2seq formulation as well as measuring the effect of the CRF layer are well taken; we will attempt to include those results in the paper and update our revision, time permitting.
>
> Experiments comparing the performance of the entailment module on the somewhat larger SciTail dataset are conducted in https://arxiv.org/pdf/1809.05724.pdf . We will update the discussion to reflect this result in the science domain, and try to provide references for its performance on other large-scale NLI corpora.
>
> In the original ARC paper (https://arxiv.org/pdf/1803.05457.pdf), they evaluate their baselines (DGEM, BiDAF, and DecompAttn) using a significance threshold of ±2.5% compare to random chance:
> “The most striking  observation is that none of the algorithms score significantly higher than the random baseline on the Challenge set, where the 95% confidence interval is ±2.5%. In contrast, their performance on the Easy set is generally between 55% and 65%. This highlights the different nature and difficulty of the Challenge set.”
> By both that measure and by using a one-sample t-test, we found that our models outperformed the reported accuracy of those three baselines at the 99% confidence interval on the test set; however, our methods are not significantly better than KG2 at the 95% level (p=0.652 for NCRF++). Comparing to the reported accuracy of the BiLSTM Max-Out model of Mihaylov et al., our regular NCRF++ models (as well as the ones augmented with TransH and PPMI embeddings) are not significantly worse at 95% confidence.
>
> On the other hand, paired t-tests comparing the AI2 rule to the Top-2 rule (without splitting) favored the Top-2 rule with at least 99.9% confidence on the test set. While we observe that splitting is harmful for the test set, the results are only significant at the 95% level for PPMI and Original Question.

---

### Official Review · AnonReviewer2 · 2019-01-08
**Good paper, some clarifications needed**

**Rating:** 7
**Confidence:** 4

**Review:**

Summary

This paper addresses the ARC dataset by reformulating the question using embeddings from ConceptNet. Their model selects a few terms from the question using the embeddings from ConceptNet, rewrites the query based on the selected terms, retrieves the documents and solves the query. The empirical result shows that embeddings from ConceptNet is beneficial, and the overall result is comparable to recent performance on ARC dataset.

Quality
pros
1) This paper contains a thorough study of recent QA models and datasets.
2) This paper describes the model architecture, conducts ablation studies of different Essential terms classification, and includes thorough comparisons with recent models on ARC challenges.

cons
- Although the paper includes recent works on QA models/datasets, it doesn’t contain much studies on query reformulations. For example,  "Ask the Right Questions: Active Question Reformulation with Reinforcement Learning” (Buck et al., ICLR 2018) is one of the related works that the paper didn’t cite.
- The paper does not have any example of reformulated queries or error analysis.

Clarity

pros
1) The paper describes the framework and model architecture carefully.

cons
1) It is hard to understand how exactly they reformulate the query based on selected terms. (I think examples would help) For example, in Fig 2, after “activities”, “used”, “conserve” and “water” were selected, how does rewriter write the query? The examples will help.
2) Similar to the above, it would be helpful to see the examples of decision rules in Section 5.2.
3) It is hard to understand how exactly each component of the model was trained. First of all, is rewrite module only trained on Essential Terms dataset (as mentioned in Section 3.1.3) and never fine-tuned on ARC dataset? Same question for entailment modules: is it only trained on SciTail, not fine-tuned on ARC dataset? How did decision rules trained? Are all the modules trained separately, and haven’t been trained jointly? What modules were trained on ARC dataset? All of these are a bit confusing since there’re many components and many datasets were used.

Originality & significance

pros
* Query reformulation methods have been used on several QA tasks (like Buck et al 2018 above), and incorporating background knowledge has been used before too (as described in the paper), but I think it’s fairly original to do both in the same time.

cons
* It is a bit disappointing that the only part using background knowledge is selecting essential terms using ConceptNet embedding. I think the term “using background knowledge” is too general term for this specific idea.

In general, I think the paper has enough contribution to be accepted, if some descriptions are better clarified.

---

> ### Author Response · Authors · 2019-02-01
> **Added discussion of AQA, example, and clarifications on training -- many good ideas for future work!**
>
> Thank you for the very considerate review!
>
> We will include a discussion of our paper in the context of the AQA work, which tackles the very interesting problem of open-vocabulary questions reformulation. Their paper illustrates the difficulty of the task, noting that “99.8% of [AQA-QR rewrites] start with the prefix What is [name]...”.  They speculate that this “might be related to the fact that virtually all answers involve names, of named entities (Micronesia) or generic concepts (pizza).” We hope to expand our work to include question and topic expansion in the near future, as we agree that making progress on open-vocabulary query reformulation in the context of non-factoid question answering is an exciting effort.
>
> We have space in the paper and have included examples of the rewritten queries in our revision.  This also goes along with your Clarity Con point 2 -- the rewritten query is a concatenation of the selected terms.
>
> We will try to work in an example of the decision rule in the appendix.  Since the decision rule uses many query results it takes up significant space.  Informally, the decision rule looks at the average or max entailment for each answer for the various query results.  This is tuned on the ARC-Challenge-Train dataset.
>
> Indeed, as you say the rewriter is trained only on the Essential Terms dataset and the entailment is only trained on the SciTail dataset.  We are in the process of transforming some of the ARC-Challenge questions to fine tune the entailment model but this work is not completed yet.  The Decision Rules are trained on 1/5th of the ARC-Training set which we can use as we have not used the ARC-Train for any other portion of the model.
>
> We agree that leveraging background knowledge is an important line of research, and that the work is this paper represents very early steps towards fully utilizing it in an open-domain QA pipeline. Future work indeed involves jointly training the modules together -- we are in the process of integrating background knowledge into the entailment model, both by a similar approach to https://arxiv.org/pdf/1809.05724.pdf and by using more sophisticated graph embeddings.

---

### Meta-Review · Area_Chair1 · 2019-02-11
**Moderate contribution of using external knowledge to improve QA in ARC dataset**

**Recommendation:** Accept (Poster)
**Confidence:** 3

**Metareview:**

This paper presents a method for finding important tokens in the question, and then use prior knowledge from conceptNet to answer questions in the ARC dataset.

Pros.
Combining finding essential terms, using domain knowledge, and textual entailment.

Cons.
None of the proposed methods are novel.
The paper combines a few components to answer questions in ARC; the intro and abstract are written a bit more general than what the paper actually does.
The paper studies different ways to incorporate concept net; the results between dev and test are not consistent. Some methods achieve better results on the dev set, but the authors use a model (which is not best in dev) to be used on the test set.

---

### Decision · Program_Chairs · 2019-02-15
**AKBC 2019 Conference Decision**

Accept